# Fatty Acid Release and Gastrointestinal Oxidation Status: Different Methods of Processing Flaxseed

**DOI:** 10.3390/foods13050784

**Published:** 2024-03-03

**Authors:** Mingkai Zhang, Yashu Chen, Hongjian Chen, Qianchun Deng

**Affiliations:** 1School of Food Science and Engineering, Wuhan Polytechnic University, Wuhan 430048, China; zhangmingkai1210@163.com; 2Oil Crops Research Institute, Chinese Academy of Agricultural Science, Wuhan 430062, China; yashuchen@sina.com; 3College of Health Science and Engineering, Hubei University, Wuhan 430062, China

**Keywords:** flaxseed processing technology, simulated digestion, lipid oxidation

## Abstract

Flaxseed has been recognized as a superfood worldwide due to its abundance of diverse functional phytochemicals and nutrients. Various studies have shown that flaxseed consumption is beneficial to human health, though methods of processing flaxseed may significantly affect the absorption and metabolism of its bioactive components. Hence, flaxseed was subjected to various processing methods including microwaving treatment, microwave-coupled dry milling, microwave-coupled wet milling, and high-pressure homogenization. In vitro digestion experiments were conducted to assess the impact of these processing techniques on the potential gastrointestinal fate of flaxseed oil. Even though more lipids were released by the flaxseed at the beginning of digestion after it was microwaved and dry-milled, the full digestion of flaxseed oil was still restricted in the intestine. In contrast, oil droplets were more evenly distributed in wet-milled flaxseed milk, and there was a greater release of fatty acids during simulated digestion (7.33 ± 0.21 μmol/mL). Interestingly, wet-milled flaxseed milk showed higher oxidative stability compared with flaxseed powder during digestion despite the larger specific surface area of its oil droplets. This study might provide insight into the choice of flaxseed processing technology for better nutrient delivery efficiency.

## 1. Introduction

Flaxseed is regarded as a superfood all over the world and is rich in alpha-linolenic acid (ALA), lignans (secoisolariciresinol bisglucoside, etc.), high-quality plant proteins, dietary fiber, and vitamins [1,2,3,4]. Thereinto, ALA is an important and characteristic nutrient, being a type of essential fatty acid, and possesses multiple biological activities, including reducing cholesterol levels, regulating blood lipids, promoting normal brain development, and slowing cognitive decline during aging [5,6]. Although the importance of ALA in health has attracted much attention, the proportion of ALA-rich foods in people’s daily diets remains relatively low. For example, a recent epidemiological study suggested that an insufficient intake of omega-3 PUFAs occurred in nearly 195 countries and regions. Additionally, a lack of omega-3 PUFAs can cause an imbalance in omega-6 PUFAs and omega-3 PUFAs in the human body, resulting in cardiovascular, cerebrovascular, and autoimmune diseases [7]. At the same time, the omega-3 PUFA in flaxseed, namely ALA, needs to be properly processed to increase its stability and bioavailability. Hence, there is an increasing demand for the development of flaxseed processing techniques aside from the traditional oil-pressing process.

Recently, developing new techniques of flaxseed processing has become a research hotspot to encourage the intake of ALA. Since microwave processing can give flaxseed a crispy texture and nutty taste [8], microwave-treated flaxseed has been increasingly consumed as an ingredient in bread, biscuits, yogurt, and other foods [9]. At the same time, flaxseed powder obtained by dry milling skim or whole-fat flaxseed is convenient to use in smoothies, dairy products, and baked goods to improve the nutritive value of the final product. Moreover, according to published articles, dry milling flaxseed can improve the taste and quality of biscuits and bread [10]. Compared to flaxseed powder, whole flaxseed has a longer shelf life with a slower oxidation rate and is easier to store. In general, flaxseed and flaxseed powder can not only be eaten directly but also within multiple fortified foods, and microwaving and dry milling are typical techniques for processing them [8].

Owing to their healthful qualities (high in unsaturated fat and polyphenol contents) and unique flavors, the consumption of various plant-based milks (e.g., oat, soybean, and almond milk) is currently growing [11,12]. They serve as alternatives to milk for consumers with lactose intolerance or religious beliefs and those who are vegetarian [13]. Flaxseed plant-based milk can be obtained by wet milling flaxseed coupled with high-pressure homogenization [14]. Due to the large specific surface area of the flaxseed oil droplets within flaxseed-based milk and readily oxidizable PUFAs, the physical and chemical stability of flaxseed-based milk in storage and the digestive tract should be modulated using an appropriate technology [15]. Notably, the oxidation products of PUFAs not only reduce the nutritional value of lipids but also increase the risk of developing cardiovascular disease, diabetes, tumors, and other pathologies after excessive consumption [16]. Although the effect of lipid oxidation products on the activation of the anti-inflammatory response has been studied [17,18], the current research still focuses on its negative effects on food quality and human health [19,20]. Meanwhile, the human gastrointestinal tract contains a variety of pro-oxidative factors such as metal ions and reactive oxygen species [21]. For example, dietary iron could be released into the stomach during the digestion of vegetables, grains, and meat [22]. Beyond that, processing techniques can severely influence the food matrix of flaxseed-fortified food, including the cell wall structure and the size of tissues, thus having a significant effect on the fate of flaxseed in the digestive tract [23,24]. Research related to flaxseed is still mainly concerned with the pharmacological properties and functionality of flaxseed itself, and less attention has been paid to the status of the nutrients in flaxseed during gastrointestinal digestion [8,25]. To date, the effects of different methods of processing flaxseed on the degree of flaxseed oil release and oxidation in digestion are relatively unexplored.

Hence, in this study, whole flaxseed, flaxseed powder, and flaxseed plant-based milk were chosen as research models to explore the effects of processing methods on the physicochemical and nutritive properties of flaxseed within the digestive tract. The manifestations of flaxseed during simulated digestion after a microwaving treatment, microwaving coupled with dry milling, microwaving coupled with wet milling, and high-pressure homogenization were thoroughly studied and discussed. The structure, particle size, and oxidation stability of these three types of samples in simulated digestion were fully determined and compared. Our results might provide important insights into novel processing techniques for flaxseed for improving its lipid oxidative stability and bioaccessibility.

## 2. Materials and Methods

### 2.1. Materials

Flaxseeds were procured from the Gansu Academy of Agriculture (Zhangye, China); the variety is Zhangya No. 2. Nile red (≥98%, CAS: 7385-67-3) was purchased from McLean (Beijing, China). All inorganic chemical reagents were purchased from Sinopharm Chemical Reagent Co., Ltd. (Beijing, China). 2-Methyl-3-heptanone (95.0%, CAS: 13019-20-0) was purchased from TCI Ltd. (Tokyo, Japan). Mucin (CAS:84082-64-4), Pepsin (CAS: 9001-75-6), Pancreatin (CAS: 8049-47-6), RGE (RGE-15), Hematin (CAS: 15489-90-4), and bile salts were purchased from Sigma (New York, NY, USA).

### 2.2. Pretreatment of Flaxseed and Preparation of Flaxseed-Based Milk

Whole flaxseeds (WFs): The water content of flaxseeds was adjusted from 3.4% to 19% at 4 °C before heating. Flaxseeds (19 g) were placed in a Petri dish with a diameter of 9.5 cm and microwaved for 5 min at 700 W (CEM Mars-6, Matthews, NC, USA). The resulting microwaved flaxseeds were referred to as whole flaxseeds (WFs) in this study. Subsequently, the product was ground in a grinder (the particle size of the sample was less than 3000 μm) (GX200, Shijiazhuang, China) for 10 s [26], simulating the human chewing process [27].

Flaxseed powder (FP): After the same microwave treatment and moisture adjustment process, the microwaved flaxseeds were ground in a grinder (GX200, Hebei, China) for 20 s. The flaxseeds after microwaving and dry milling were referred to as flaxseed powder (FP) in the study.

Flaxseed-based plant milk (FM): Microwaved flaxseeds were soaked in water at 25 °C for 2 h (1: 7, *w*/*v*). The soaked flaxseeds were mixed with deionized water (1: 7, *w*/*v*; pH = 7.0) and ground using a colloid mill (Horizontal-60, Shenzhen, China). After that, 1% cellulase and 2% glucoamylase were added to the sample for enzymolysis at 50 °C for 1 h and then it was placed at 95 °C for 15 s to inactivate the enzyme. The resulting emulsion was filtered through a 200-mesh filter cloth and homogenized under a pressure of 20 Mpa (GYB40-10S, Shanghai, China). Finally, flaxseed-based plant milk (FM), the wet-milled flaxseed, was obtained by sterilization at 137 °C for 15 s (ST21-4338-1, Shanghai, China).

After all samples were packed with nitrogen, they were placed at −20 °C and the follow-up simulated digestion experiments were completed within 1 week.

### 2.3. In Vitro Simulated Digestion

Based on INFOGEST 2019 [28], an in vitro semi-dynamic digestion model was used to simulate the digestion of WF, FP, and FM. For oral digestion, 20 g samples were gently mixed with 20 mL simulated salivary fluid (SSF) at 37 °C for 2 min using a thermostatic shaking water bath at 100 rpm (Shanghai Boxun, SHZ-B, Shanghai, China).

A total of 20 g of this portion of digesta was then immediately transferred to a water-packed glass container containing 10% simulated gastric fluid (SGF) to simulate the basal volume of an empty stomach. This vessel was connected to a thermostatic water circulator (Mettler Toledo, Greifensee, Switzerland) to adjust the temperature to 37 °C before being placed on an orbital shaker (Servicebio, Wuhan, China) at 20 rpm for gentle mixing of the vessel contents. Hematin was the main source of total iron in the body; the concentration of heme iron in gastric fluid was approximately 20 μM [29]. Based on the INFOGEST model, heme iron (20 μm in methanol) and 60 U/mL RGE-15 was added to the stomach portion of the simulated digestive system. The remaining 90% SGF was then added and Milli-Q water was added until the whole sample weighed 40 g, shaking for 2 h to perform simulated stomach digestion.

The small intestine environment was simulated in vitro using a pH Stat (Mettler Toledo, Greifensee, Switzerland). During the simulation of digestion in the small intestine, 0.1 M NaOH solution was dripped continuously to maintain the pH of the whole reaction system at 7.0. The whole intestinal digestion stage took 2 h. The amount of released free fatty acids (FFAs) was calculated using the recorded volume of NaOH required for neutralization.
$$FFA=\frac{V_{NaOH}\times C_{NaOH}\times 1000}{V_{intestinal\;digestion\;mixture}}$$
where V_NaOH_ and C_NaOH_ represent the consumed volume and molar content of the NaOH, respectively, and V_intestinal digestion mixture_ represents the volume of the mixed intestinal digestion fluid [30].

### 2.4. Particle Size Measurement

The particle size distributions and mean particle sizes of WF, FP, FM, and their digesta (expressed as D[4,3]) were measured during digestion using a Mastersizer 3000 static light scatterer (Malvern Instruments, Malvern, UK). Milli-Q water was selected as the dispersant for the original samples. For the digested samples, the dispersant used was the digestion fluid from each digestion stage [31]. Since WF and FP are solid particles, determining their original sample size requires diluting them in Milli-Q water to maintain the same lipid content as that of the original FM samples.

### 2.5. Lipid Content Determination

The WF, FP, and their chyme at different stages of digestion were collected and naturally dried at 25 °C. The lipid contents of the samples were determined using a Soxhlet extraction apparatus with petroleum ether as the solvent. The results of the lipid content analysis were expressed as a percentage relative to the dry weight [32].

### 2.6. Observation of Microstructure

Briefly, 200 μL of the original and digested samples was diluted with deionized water and simulated digestion fluid from each stage in a 1:1 (*v*/*v*) ratio. Additionally, 10 μL of Nile red at a concentration of 1 mg/mL was added for staining purposes [31]. A small portion of softened digesta obtained from the WF and FP using tweezers was carefully placed on a slide. The distribution of oil droplets in the samples was observed using a laser scanning confocal microscope (LSCM) (Zeiss, LSM980, Chiba, Japan).

### 2.7. Determination of Volatile Unsaturated Aldehydes in Secondary Oxidation Products

The 20 mL headspace flasks were used to weigh 5 g of WF, FP, FM, and their digesta at each stage of digestion. An internal standard of 1 μL 2-methyl-3-heptanone was added before equilibrating the samples at 40 °C for 20 min. The headspace compounds were then absorbed by DVB/CAR/PDMS fibers for 50 min and injected into the injection port for a duration of 5 min. A 7890A gas chromatograph (Agilent, Santa Clara, CA, USA) was used in combination with a 5975C mass spectrometer (Agilent, Santa Clara, CA, USA) to detect the volatile compounds. Separation occurred on DB-WAX (30 m × 0.25 mm × 0.25 μm) columns with helium as the carrier gas at a flow rate of 1.5 mL/min and an injection temperature set to 250 °C. The initial temperature was 40 °C; this was held for 2 min before heating to 180 °C at 5 °C min^−1^. This temperature was held for 2 min before heating to 240 °C at 8 °C min^−1^. The ion source temperature, electron energy, transmission line temperature, and mass scanning range were 230 °C, 70 eV, 280 °C, and 40–350 *m/z*, respectively [26].

### 2.8. Data Analysis

Data were presented as mean ± SD (*n* = 3). A one-way ANOVA, followed by Tukey’s test, was performed to analyze the significant differences between the data (*p* < 0.05) using SPSS 24 (SPSS Inc., Chicago, IL, USA).

## 3. Results and Discussion

### 3.1. Changes in FM, FP, and WF during Oral In Vitro Digestion

The particle sizes of the WF, FP, FM, and their digesta are characterized and presented in Figure 1. The results revealed distinct patterns in the particle size distribution of these three samples during the oral digestion stage. Due to the limited duration of this stage, the simulated oral digestive fluid had a negligible impact on the particle size distribution of WF and FP, which can be attributed to their gradual release of lipids, proteins, and other substances. On the other hand, the particle size distribution diagram showed that the small peak below 1 μm of the FM vanished and a new peak above 10 μm appeared, suggesting that oil droplets within FM flocculated due to mucin in the oral digestive fluid [33].

CLSM was employed to observe the alterations in microscopic states of the WF, FP, and FM during the entire process of digestion. Confocal micrographs of FM during the oral digestive stage (Figure 2a) revealed a more homogeneous state within the entire system due to its delicate structure encompassing oil bodies, colloidal macromolecules, etc. [14]. Hence, during the oral stage of digestion, the particle size D[4,3] in the FM showed only relatively minor changes (Figure 1a and Figure 2a). Given that solid particles in WF and FP were not uniformly distributed during the digestion process, separate observations were made on digestive fluid and chyme in WF and FP to examine nutrient distribution across the entire system as well as lipid release extent from solid particles (Figure 2b,c).

As shown in Figure 2b,c, in the oral digestive fluid, unevenly distributed oil droplets were released into the oral digesta from WF and FP, and oil droplets clustered significantly. The CLSM analysis of the chyme revealed ruptured edges of flaxseed tissues, leading to a partial release of lipids. In contrast, the core of the solid particles in the WF and FP were not successfully stained, indicating that lipids located within the interior of flaxseed tissues were sealed during the oral digestion stage. These oil droplets combined with other substances such as flaxseed protein might need further enzymolysis to be released in the subsequent digestive process. Precisely because lipid residues were observed in solid granules of FP and WF after oral digestion, the lipid contents of FP and WF remaining in the granules after different stages of digestion were determined (Figure 3). The percentages of lipids remaining in FP and WF after oral digestion were 74.27 ± 0.40% and 84.44 ± 0.43%, respectively.

The volatile unsaturated aldehydes produced by these three samples during digestion were analyzed using gas chromatography–mass spectroscopy (GC-MS). Additionally, the content of volatile aldehydes generated by each sample at different stages of digestion was quantified and expressed as ng/g oil, based on the lipid release data shown in Figure 3. In the initial state, only a few types and low amounts of oxidation products were observed (Table 1). Similarly, the peroxide value of the samples before the experiment was tested, but the value obtained did not exceed the effective detection limit (data not shown in the article), indicating low oxidation levels in the samples prior to digestion. Due to the short duration of the oral digestion, the types of oxidation products produced at this stage were similar to the original samples. According to the above results, the influence of three different processing methods, including microwaving treatment, dry milling, and wet milling, both on the original and oral oxidation states of flaxseed, was negligible.

### 3.2. Changes of FM, FP, and WF during Gastric In Vitro Digestion

After undergoing gastric digestion, the average size of FM particles (D[4,3]) increased from 5.58 ± 0.16 μm to 32.93 ± 6.43 μm (Figure 1). One possible explanation for this phenomenon could be attributed to the hydrolysis of proteins at the oil droplet interface by proteases present in gastric digestive fluid, which significantly impacted the stability of FM and led to aggregation or coalescence of the oil droplets [34]. According to Figure 1, during the gastric digestion stage, the particle sizes (D[4,3]) of the WF and FP decreased from 1086.67 ± 24.94 and 497.33 ± 20.34 μm to 473.33 ± 28.11 and 121.33 ± 5.44 μm, respectively. After gastric digestion, the size of solid particles, such as almond and other nuts, could decrease significantly [23]. The differences between the CLSM results of FP and WF during the gastric digestive stage were not obvious (Figure 2b,c). The lipid particle sizes were reduced in the digestive fluid, but still much larger than that in the FM, while the structures of the particle edges were further disrupted. Moreover, fewer lipids remained at the edges of the food chyme, with more fluorescence staining inside the particles. The percentages of remaining lipids in the two samples were 53.89 ± 0.33% and 71.84 ± 0.28% after gastric digestion, respectively (Figure 3). The above results indicated that the nutrients of FP and WF in gastric digestive fluid were further released and digested owing to the presence of gastric lipases and proteases [27], but there were still a lot of lipids remaining in the solid particles. The particles in the chyme were continuously broken in the gastric environment and the lipids were gradually released at the edges. At the same time, the mechanical action from the stomach alone was not enough to fully degrade the cell wall within the core of the granule, and as a result, the cores of the particles were still not all stained [27].

With the increase in digestion time, the content of various long-carbon-chain unsaturated aldehydes was increased in the gastric digestive stage (Table 1), which exhibited a positive correlation with the degree of lipid oxidation [35]. From the initial stage of lipid oxidation, with the extension of oxidation time, the concentration of hydrogen peroxide will first increase and then decrease. The total concentration of volatile oxidation products will continue to rise throughout the oxidation process. Then, when the concentration of volatile products increases, the hydrogen peroxide concentration may still be at a high value or will show a downward trend. It is undeniable that as the concentration of hydroperoxides increases, the concentration of volatile products also increases [29]. The oxidation degree and volatile unsaturated aldehydes produced by FM, FP, and WF during gastrointestinal digestion were comprehensively compared using a PCA model [35]. Due to the highest levels of (E,E)-2,4-heptadienal, (E)-2-heptadienal, and (E)-2-pentadienal, FP was placed on the far right of the score plot (Figure 4B). Additionally, in Figure 4A, both FP and WF were classified in the lower half of the score plot due to the generation of hexanal. In Figure 4A, it could be seen that the oxidation degree of FP was higher than that of WF and FM. Meanwhile, according to Table 1, in the gastric stage, the types and contents of aldehydes produced from FP were higher than FM, which is rather interesting considering the higher specific surface area of flaxseed oil droplets of FM. Even though the lipids in the FM were more exposed to the digestive fluid, the stabilization of amphipathic proteins and the adsorption of antioxidant substances at the interface resulted in certain antioxidant properties [36,37]. In contrast, the released lipids of FP in the digestive fluid were not uniformly encapsulated, resulting in increased lipid oxidation [38]. Overall, short-carbon-chain aldehydes were produced more by WF and FP in simulated gastric digestion, presumably due to the collapse of the oil bodies in solid particles, which led to triglyceride spillover and inconsistent spatial distribution of antioxidant substances with triglycerides [39,40,41].

### 3.3. Changes in FM, FP, and WF during Intestinal In Vitro Digestion

During intestinal digestion, all the samples exhibited rapid release of FFAs in the early stages and slow release during the later stages. Compared with WF and FP, the total amount of fatty acids released by FM was the largest (7.53 ± 0.21 μmol/mL) (Figure 5A). Moreover, none of the samples reached a plateau after 120 min due to long-chain triacylglycerol digestion [42]. The amount of fatty acids released was affected by the composition and structure of the interface and the droplet size [43]. The rapid FFA release was likely caused by small droplets with larger oil–water interfacial area undergoing rapid lipolysis [44]. In addition, the faster release rate of FFA in the early stages might be attributed to the faster adsorption rate of lipase on the oil droplet surface [45]. Meanwhile, the cell walls of the WF and FP presented a greater barrier to lipase. Larger particle size made it much more difficult for there to be contact between lipids and digestive fluid [46]. After wet milling, the nutrients in flaxseeds were dispersed to the maximum extent in water. The highest degree of lipid hydrolysis was observed in FM, followed by FP and WF, resulting in the release of 26.47 ± 0.21% and 34.03 ± 0.25% residual lipids in solid particles of FP and WF after intestinal digestion (Figure 3). During both the oral and gastric digestive stages, WF exhibited lower oil release compared to other samples, with a significant amount of lipids being released only during intestinal digestion. These findings indicated that the cell structure in WF granules remained highly stable during gastric digestion, resulting in a reduction in cell fragmentation [27]. Furthermore, compared with FM and FP, WF had the lowest concentration of fatty acids in the intestinal digestive fluid (Figure 5B). At the same time, FM and FP had similar fatty acid release efficiency and FM exhibited a higher total amount of fatty acid release. Overall, these results highlight the significant impact of different processing techniques on flaxseed lipid digestion. Specifically, FM prepared by microwaving coupled with wet milling and high-pressure homogenization could significantly enhance lipid digestion compared to dry milling.

During gastrointestinal digestion, the particle size of FP decreased more significantly than that of WF (Figure 1). This could be attributed to the fact that smaller solid food particles tend to undergo greater fragmentation during digestion [27]. In contrast, the average size of FM particles (D[4,3]) decreased from 32.93 ± 6.43 μm to 9.73 ± 0.44 μm after small intestinal digestion, and there was a significant increase in the proportion of particles with a size less than 1 μm. This suggested that the oil droplets in FM had a larger surface area during intestinal digestion. More sites on the interface could increase the contact between lipids and pancreatic lipase, resulting in more lipid hydrolysates and formation of absorbable micelles [47,48]. In intestinal digestion, the degree of fragmentation of solid particles within the chyme in the FP and WF increased but was not significantly different from that of those in the gastric stage (Figure 2). The structure of the particles was continuously broken and disintegrated by proteases present in the digestive fluid, while tissue fragmentation during intestinal digestion was not prominently evident [49]. However, most of the lipids inside the cells were stained and almost no oil droplets remained at their edges. At this point, the lipid content in the digestive fluid of both FP and WF was very low, similar to FM. The confocal micrographs of the WF exhibited minimal fluorescence signal. Due to the influence of bile salts on both oil droplet and cell structure, pancreatic lipase interacted with free oil droplets as well as lipids at the edges and core of the granules. Similar to the FM, lipids released by the WF and FP into the digestive fluid were rapidly digested. However, during digestion, the release of lipids inside the solid particles could be significantly blocked due to the intact cellular structure [49,50]. More oil droplets of the FM were released into the digestive fluid, leading to a higher fatty acid release rate. In contrast, the FP displayed a more disrupted structure and less intact internal cells than the WF, thus releasing more lipids into the digestive fluid. Nevertheless, a large amount of lipids remained within the cells, resulting in lower fatty acid release compared to that observed for FM (Figure 5).

Lipids in the digestive fluid were gradually hydrolyzed into fatty acids through the synergistic action of pancreatic lipase and bile salts within the intestine. At the same time, the total amount of unsaturated aldehydes produced by FM, FP, and WF decreased in intestinal digestion compared to gastric digestion. This phenomenon may be because the reaction substrate to produce a certain volatile product had a structural change due to oxidation, which made it easier to generate more complex volatile oxidation products. At the same time, these volatile products with more complex structures were not easily separated by detection instruments, making these products difficult to detect. Therefore, the reduction in detected concentrations of volatile unsaturated aldehydes at later stages of lipid oxidation does not necessarily indicate a diminished extent of lipid oxidation [35]. In the intestinal stage, more hexanal and (E)-2-pentenal were produced by FM, while more (E,E)-2,4-Heptadienal was produced by FP (Table 1). Due to the formation of (E,E)-2,4-Heptadienal, hexanal, and (E)-2-pentenal, FM and FP were classified in the lower half of the score plot (Figure 6). Compared with FP and WF, more kinds of volatile unsaturated aldehydes were released by FM at this stage. In view of the fact that the least amount of lipids was released and hydrolyzed from WF, the amount of aldehydes generated following intestinal digestion was reduced [47]. Overall, in the simulation of digestion, FP contained more long-carbon-chain unsaturated aldehydes; thus, it might have a higher lipid oxidation degree than FM.

## 4. Conclusions

In general, at the beginning of digestion, the distribution of flaxseed oil in FM was more uniform and the whole system was more stable due to wet milling coupled with high-pressure homogenization. Subsequently, during the later stages of digestion, hydrolysis of the oil droplet interface by digestive enzymes led to complete lipid hydrolysis and subsequent release of free fatty acids. However, it was observed that the cell structure within WF and FP did not fully disrupt the digestion process, resulting in a lower rate of fatty acid release. Owing to the high degree of fragmentation and small granule size, more lipids were released from FP into the digestive fluid. Simultaneously, the lipid oxidation degree of FP increased during digestion due to the larger oil–water interface without protection. In contrast, the lipids in the WF were oxidized at a lower rate during digestion, which might be due to the low degree of granule breakage and greater restriction on the chain reaction of oil oxidation. Overall, more fatty acids were released by FM than FP during digestion, while the total oxidation level of lipids in FM was lower than that in FP. Different processing methods of flaxseed significantly influence the digestion and absorption of long-chain polyunsaturated fatty acids, including their oxidative stability.

## Figures and Tables

**Figure 1 foods-13-00784-f001:**
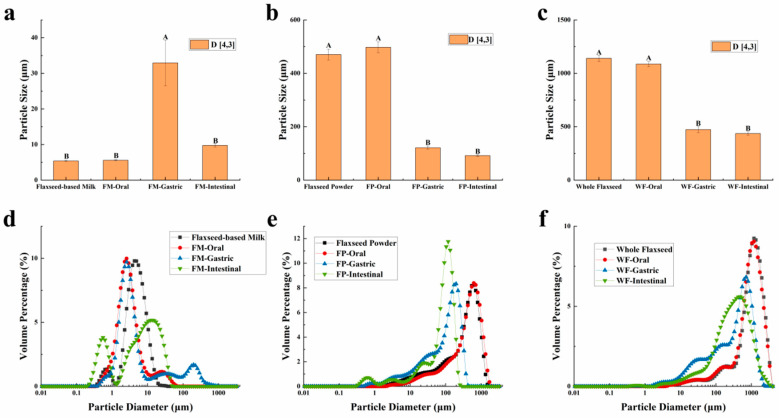
(**a**,**d**) The mean particle diameter and distribution of flaxseed-based milk (FM), (**b**,**e**) flaxseed powder (FP), (**c**,**f**) whole flaxseed (WF), and their digesta after each stage of simulated digestion. D[4,3] was used to evaluate the mean particle size of the samples. Data points and error bars represent means (*n* = 3) ± standard deviations. Different letters between groups indicate significant differences between samples (*p* < 0.05).

**Figure 2 foods-13-00784-f002:**
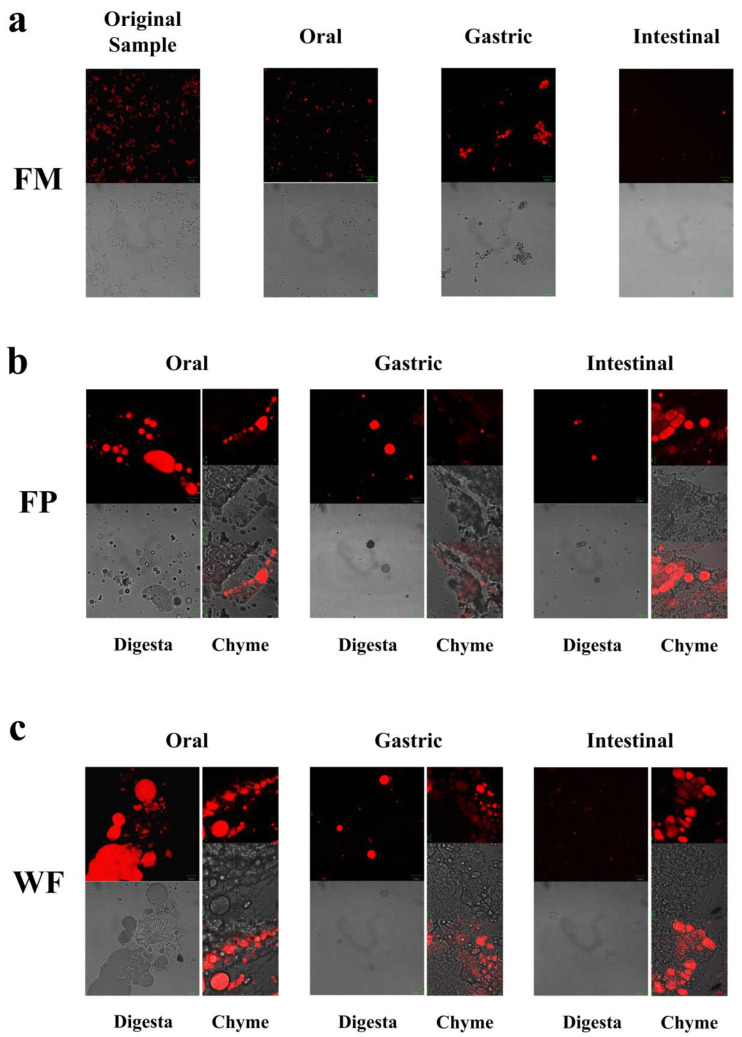
(**a**) Microstructures of flaxseed-based milk (FM), (**b**) flaxseed powder (FP), (**c**) whole flaxseed (WF), and their digesta at various digestive stages.

**Figure 3 foods-13-00784-f003:**
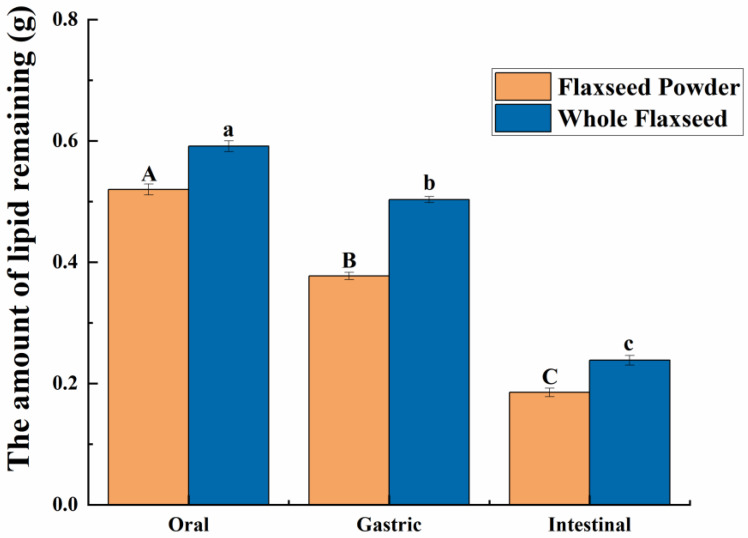
The amount of lipids remaining due to the flaxseed powder (FP) and whole flaxseed (WF) during each stage of simulated digestion. Different letters between groups indicate significant differences between samples (*p* < 0.05). The significance of the data differences between different groups is distinguished by uppercase and lowercase letters.

**Figure 4 foods-13-00784-f004:**
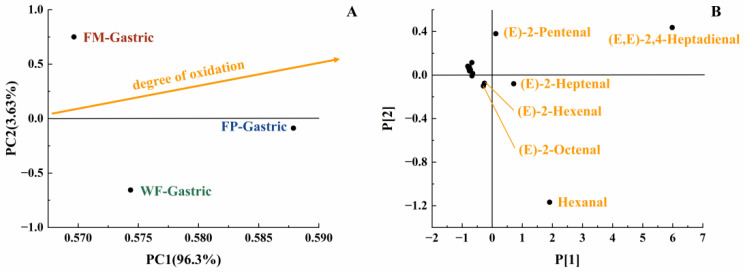
PCA model of GCMS data of gastric digestive stage (PC1 vs. PC2; R^2^X[1] = 0.963, Q^2^[1] = 0.577; R^2^X[2] = 0.036, Q^2^[2] = 0.002) with scores plot (**A**) and loadings plot (**B**).

**Figure 5 foods-13-00784-f005:**
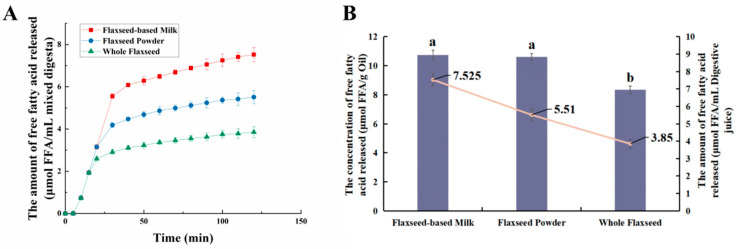
(**A**) Trends of fatty acid release from whole flaxseed (WF), flaxseed powder (FP), and flaxseed-based milk (FM) during simulated digestion in the intestine. (**B**) The concentration and amount of fatty acids released by flaxseed-based milk (FM), flaxseed powder (FP), and whole flaxseed (WF) in simulated digestion. Data points and error bars represent means (*n* = 3) ± standard deviations. Different letters between groups indicate significant differences between samples (*p* < 0.05).

**Figure 6 foods-13-00784-f006:**
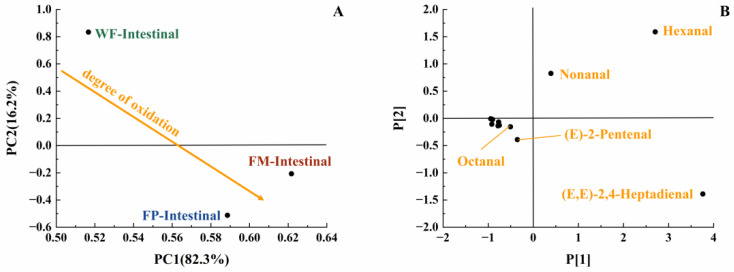
PCA model of GCMS data of intestinal digestive stage (PC1 vs. PC2; R^2^X[1] = 0.823, Q^2^[1] = 0.575; R^2^X[2] = 0.162, Q^2^[2] = 0.038) with scores plot (**A**) and loadings plot (**B**).

**Table 1 foods-13-00784-t001:** Volatile unsaturated aldehyde content generated by flaxseed-based milk (FM), flaxseed powder (FP), whole flaxseed (WF), and their chymes at different stages of simulated digestion. Different letters following the mean in the same column indicate significant differences (*p* < 0.05).

Phase	Name of Compound	FM (ng/g Oil)	FP (ng/g Oil)	WF (ng/g Oil)
Original Sample	Hexanal	21.1 ± 4.1 ^b^	34.1 ± 2.9 ^b^	58.8 ± 10.4 ^a^
	Nonanal	-	-	19.9 ± 3.4
Oral	Hexanal	34.7 ± 2.5 ^b^	46.3 ± 2.8 ^b^	71.8 ± 2.2 ^a^
	Octanal	4.5 ± 0.3	-	-
	Nonanal	7.9 ± 0.8	-	38.0 ± 8.7
Gastric	Propanal	-	17.3 ± 2.2	17.8 ± 0.1
	Pentanal	-	22.5 ± 0.8	24.0 ± 1.7
	2-Butenal	-	55.2 ± 1.3	52.9 ± 3.6
	Hexanal	166.0 ± 7.3 ^c^	663.6 ± 43.6 ^b^	865.7 ± 40.8 ^a^
	(E)-2-Pentenal	213.2 ± 9.0 ^a^	166.3 ± 11.3 ^b^	155.1 ± 7.7 ^b^
	3-Hexenal	-	18.8 ± 3.7	0.018.0 ± 2.5
	Heptanal	6.2 ± 2.1 ^b^	27.6 ± 0.4 ^a^	35.2 ± 4.9 ^a^
	(E)-2-Hexenal	56.6 ± 8.1 ^b^	126.7 ± 0.7 ^a^	167.5 ± 8.5 ^a^
	Octanal	-	22.7 ± 1.4	32.0 ± 2.9
	(E)-2-Heptenal	205.2 ± 23.8 ^b^	341.9 ± 2.0 ^a^	371.5 ± 21.0 ^a^
	Nonanal	12.8 ± 0.7 ^c^	39.8 ± 2.8 ^b^	61.5 ± 10.0 ^a^
	(E,E)-2,4-Hexadienal	-	42.9 ± 11.1	10.0 ± 13.8
	(E)-2-Octenal	42.1 ± 7.4 ^b^	132.3 ± 27.8 ^a^	158.1 ± 10.7 ^a^
	(E,E)-2,4-Heptadienal	1101.1 ± 29.0 ^c^	1556.9 ± 176.8 ^a^	1315.8 ± 24.6 ^b^
	Benzaldehyde	28.0 ± 6.3 ^b^	33.1 ± 5.1 ^a^	31.5 ± 3.9 ^a^
	2-Decenal	-	15.0 ± 1.9	17.1 ± 5.0
	(E,E)-2,4-Decadienal	-	7.0 ± 0.9	10.6 ± 1.4
Intestinal	Hexanal	36.8 ± 2.4 ^a^	33.9 ± 0.6 ^b^	19.9 ± 2.1 ^c^
	(E)-2-Pentenal	10.8 ± 0.6	14.8 ± 0.2	-
	Heptanal	8.8 ± 0.1	-	-
	Octanal	16.0 ± 2.2	-	-
	(E)-2-Hexenal	-	7.4 ± 0.2	-
	(E)-2-Heptenal	4.6 ± 0.7	6.0 ± 0.9	-
	Nonanal	15.9 ± 0.8 ^a^	8.2 ± 0.2 ^b^	8.5 ± 0.8 ^b^
	(E,E)-2,4-Heptadienal	58.7 ± 10.8 ^b^	88.1 ± 3.2 ^a^	7.9 ± 0.2 ^c^
	Decanal	4.2 ± 0.3	-	-
	Benzaldehyde	6.4 ± 0.6	4.5 ± 0.4	-
	(E)-2-Nonenal	4.0 ± 0.5	-	-
	(E,E)-2,6-Nonadienal	5.1 ± 0.4	-	-

## Data Availability

The original contributions presented in the study are included in the article, further inquiries can be directed to the corresponding author.

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
