# Peer review of "Fatty Acid Release and Gastrointestinal Oxidation Status: Different Methods of Processing Flaxseed"

_foods, 2024, doi:10.3390/foods13050784_

Round 1

Reviewer 1 Report

Comments and Suggestions for Authors

Please elaborate on the plant species of which flaxseeds are obtained. Moreover, how did the authors confirm that all of the seeds are originated from the same species/cultivar?

The choice of the ANOVA followed by Tukey HSD is acceptable for the statistical analysis of the data.

Please highlight the novelty of this work in the abstract and Introduction.

2.5: samples were dried. Could the drying process affect the parameters measured, such as lipid content?

Author Response

Response to Reviewer 1 Comments

1.Summary

Thank you very much for taking the time to review this manuscript. Please find the detailed responses below, along with the corresponding revisions highlighted in the resubmission.

2.Point-by-point response to Comments and Suggestions for Authors

Comments 1: Please elaborate on the plant species of which flaxseeds are obtained. Moreover, how did the authors confirm that all of the seeds are originated from the same species/cultivar?

Response 1: Thank you for pointing this out. We have added the species of flaxseed in the text, which can be found on line 89 of the manuscript. We first processed flaxseed to get flaxseed milk, and then made microwaving flaxseed and dry-milling flaxseed from the raw material of flaxseed milk, so that all samples were obtained from the same raw material.

Comments 2: Please highlight the novelty of this work in the abstract and Introduction.

Response 2: Thank you for your kindly suggestion.. Therefore, we have made changes in the manuscript to emphasize this point. The revision can be found on line 75 of the manuscript. 

Comments 3: 2.5: samples were dried. Could the drying process affect the parameters measured, such as lipid content?

Response 3: Thank you for pointing this out. We deliberately chose a lower temperature (25℃) to dry the samples, in order to protect the integrity of the sample structure while removing water to obtain the most accurate lipid content parameters of the samples. Moreover, since we only need to obtain the data of the lipid content of the sample at this time, we do not need to pay attention to the lipid oxidation caused by drying treatment.

Reviewer 2 Report

Comments and Suggestions for Authors

The original research by Zhang et al. is an exciting and well-experimental design.

-The abstract provides clear information and is attractive to the reader.

-The introductions provide significant information related to the study's aims.

-The methods and results are also well done with quality.

General concept comments

-This study provides information on technology that improved flaxseed processing for better nutrient delivery efficiency and will be beneficial to human health.

The English language is well done.

Author Response

Thank you very much for taking the time to review this manuscript. Thank you for your comments on this article.

Reviewer 3 Report

Comments and Suggestions for Authors

This paper describes the effect of using different forms of flaxseed (whole flaxseed, flaxseed powder, and flaxseed-based milk), processed with defferent techniques, on the breakdown of fatty acids in the gastrointestinal tract using in-vitro digestion.The paper is well-designed and well-written, and it deals with a very interesting field. I believe it will be of interest to the readers of this journal. However, there are a few issues that need to be addressed:

1. The novelty of the study should be emphasized.

2. In line 70, change "mothed" to "method."

3. In line 224, you mentioned "According to the published articles [29]" but only referred to one. It would be better to rephrase this.

4. In line 226, "According to figures 1," I understand what you mean, but it should be written as "Figure."

Author Response

1.Summary

Thank you very much for taking the time to review this manuscript. Please find the detailed responses below, along with the corresponding revisions highlighted in the resubmission.

2.Point-by-point response to Comments and Suggestions for Authors

Comments 1: The novelty of the study should be emphasized.

Response 1: Thank you for your professional guidance, we agree with that. Therefore, we have made changes in the manuscript to emphasize this point. The revision can be found on line 75 of the manuscript.

Comments 2: In line 70, change "mothed" to "method." In line 224, you mentioned "According to the published articles [29]" but only referred to one. It would be better to rephrase this. In line 226, "According to figures 1," I understand what you mean, but it should be written as "Figure."

Response 2: Thank you for pointing this out. These phrases have been modified by us in the manuscript. The revision can be found on line 80, 244 and 246 of the manuscript. 

Reviewer 4 Report

Comments and Suggestions for Authors

The manuscript "Fatty Acid Rele ase and Gastrointestinal Oxidation Status: 2 Different Processing Methods of Flaxseed" has some strong points and should be further considered for publication after some major revisions. Please see the attached pdf file for my comments.

Comments on the Quality of English Language

moderate English editing is required

Author Response

1.Summary

Thank you very much for taking the time to review this manuscript. Please find the detailed responses below, along with the corresponding revisions highlighted in the resubmission.

2.Point-by-point response to Comments and Suggestions for Authors

Comments 1: volatiles are products of secondary oxidation. What happens with hydroperoxides? research has shown that you might have low secondary products but high primary and therefore your sample is still oxidized. Specially in the gastric phase, peroxides can be much higher that secondary products. Please discuss more extensively the interlationship between free fatty acid release, oxidation and bioaccessiblity....

Response 1: Thank you for pointing this out. We agree with what you said, because lipid oxidation is a chain reaction, there are more primary oxidation products and fewer secondary oxidation products in the stomach, so we added relevant explanations in the original article. The fact that we refer to a reduction in the total amount of volatile products at the stage of intestinal digestion does not mean that we assume that lipids are less oxidized at this stage of digestion. Similarly, in the later stages of lipid oxidation, an increase in the content of secondary oxidation products is accompanied by a decrease in the content of hydroperoxide. The changes of volatile products can also reflect the changes of hydroperoxide to a certain extent. So we added to the manuscript the analysis of the degree of lipid oxidation of the samples during digestion. The concentration of hydroperoxide is usually determined by the peroxide value method, which is a common color reaction used to reflect the degree of oxidation of the sample. Meanwhile, lipid oxidation reaction is a chain reaction process triggered by free radicals. Relatively speaking, the sensitivity of color reaction is not so high. Volatile oxidation products have been mentioned many times in the relevant studies of lipid oxidation. The precision of volatile products detection and analysis by GCMS is higher, and it can reflect the change of oil oxidation degree at later stage. The GC-MS detection method we used can avoid the samples being processed for too long and can also obtain more information to analyze the oxidation status of lipids in the samples. This is why we chose HS-SPME-GCMS to detect the oxidation products produced by the sample. Similarly, many lipid oxidation studies also focus on volatile oxidation products. For example, https://doi.org/10.1016/j.foodchem.2022.132882, https://doi.org/10.1016/j.foodchem.2020.127148 and https://doi.org/10.1002/jsfa.11904. This also shows that GCMS is sufficient for the detection of oxidation products from lipids. Therefore, we mainly discuss volatile oxidation products in the manuscript. At the same time, in accordance with your comments, we have added a discussion of the relationship between free fatty acid release, oxidation and bioavailability to the manuscript.

3. Response to Comments on the Quality of English Language

Point 1: In line 14, "were" should be changed to "was".

Response 1:We have revised this point, as can be seen in 14 lines of the manuscript. 

Point 2: In line 18, "intestinal" should be changed to "intestine" or "intestinal phase".

Response 2:We have revised this point, as can be seen in 20 lines of the manuscript. 

Point 3: In line 71, "withing" should be changed to "within".

Response 3:We have revised this point, as can be seen in 81 lines of the manuscript.

Point 4: In line 171, "variation rules" should be changed.

Response 4:We have revised this point, as can be seen in 187 lines of the manuscript.

4. Additional clarifications

We have made corresponding modifications to the other questions you raised in the manuscript, please see the attachment for details. Thanks again for your comments on this article.

Reviewer 5 Report

Comments and Suggestions for Authors

I have two points of concern that I encountered when reading the manuscript. One concerns the description of the in vitro digestion. Please clarify the methodology and describe the sampling technique. In the results, the following sentences are given - with increasing gastric digestion time .... (gastric phase length not described).

In addition, the manuscript title encompasses diverse processing methods. The study's aim is to provide an explanation of the impact of processing methods on linseed's in vitro digestibility. The results are not evaluated in relation to the impact of processing methods on the digestion of flaxeseed samples.

I am against using the word milk to refer to almond milk, plant milk, flaxeseed milk, etc. as milk is produced by the mammary glands.

My suggestion is to revise the keywords, as well as one of the keywords provided in the manuscript the bioaccessibility of w-3 polyunsaturated fatty acids has not been studied.

The methodology states that a one-way ANOVA test was used to analyze significant differences between data (p<0.05). However, the results were not analysed using a p-value.

Clarification of the study's rationale is necessary. Additional comments are given in the manuscript.

Author Response

Summary

Thank you very much for taking the time to review this manuscript. Please find the detailed responses below, along with the corresponding revisions highlighted in the resubmission.

2.Point-by-point response to Comments and Suggestions for Authors

Comments 1: One concerns the description of the in vitro digestion. Please clarify the methodology and describe the sampling technique. In the results, the following sentences are given - with increasing gastric digestion time .... (gastric phase length not described).

Response 1: Thank you for pointing this out. We have added a description of in vitro digestion methods to the manuscript. The relevant revisions can be found on line 120.

Comments 2: In addition, the manuscript title encompasses diverse processing methods. The study's aim is to provide an explanation of the impact of processing methods on linseed's in vitro digestibility. The results are not evaluated in relation to the impact of processing methods on the digestion of flaxeseed samples.

Response 2: Thank you for your kindly suggestion. We mentioned in the article that we use many methods to process flaxseed, processing raw flaxseed into three forms of products (microwave flaxseed, flaxseed meal, flaxseed plant-based milk). Throughout the literature, we analyzed the changes in the microstructure and lipid oxidation degree of these three products during gastrointestinal digestion. These changes also reflect the impact of these processing methods on the digestion of flaxseed in the human body. Therefore, the analysis of the three samples in our manuscript is actually to analyze the effect of processing methods on the state of flaxseed in simulated digestion.

Comments 3: I am against using the word milk to refer to almond milk, plant milk, flaxeseed milk, etc. as milk is produced by the mammary glands.

Response 3: Thank you for pointing this out. In addition to the word milk, which means the milk produced by the mammary gland, it also means the white juice produced by the plant. The use of "milk" to refer to the white juice made from plant materials such as almonds and flaxseeds is common in the literature. For example, https://doi.org/10.3390/foods12193571 and  https://doi.org/10.1080/10408398.2012.761950.

Comments 4My suggestion is to revise the keywords, as well as one of the keywords provided in the manuscript the bioaccessibility of w-3 polyunsaturated fatty acids has not been studied.

Response 4: Thank you for your kindly suggestion. We agree with this point, which has been revised in the manuscript. The revisions can be found on line 26 of the article.

Comments 5: The methodology states that a one-way ANOVA test was used to analyze significant differences between data (p<0.05). However, the results were not analysed using a p-value.

Response 5: Thank you for pointing this out. We added the annotation method of significance analysis to the relevant pictures and data.

Comments 6: In Table 1, you have mentioned hexanal and other aldehydes twicely. Please clarify.

Response 6: Thank you for your kindly suggestion. In Table 1, we examined the production of volatile aldehydes by samples at various stages of the digestion process, including the original sample. So the problem with the repetition of nouns that you're referring to is actually that these substances are produced at different stages of digestion. We understand your confusion about the parameters in the table, this is a typographical issue. We have retypeset it.

Comments 7: The information given in line 235 of the text is already given in Figure 3.

Response 7: Thank you for pointing this out. In Figure 3, we measured the amount of lipid remaining in WF and FP after each stage of digestion. The data given in line 235 is the percentage change in the lipid content of these two samples after digestion in the stomach. We point this out specifically in order to make it easier for the reader to understand the amount of lipid loss of the samples after digestion in the stomach.

Comments 8: Could you clarify line 247 in the article.

Response 8: Thank you for your kindly suggestion. After simulated digestion, lipids in flaxseed will produce a variety of oxidation products, among which unsaturated volatile aldehydes are more representative. We used HS-SPME-GCMS to detect the volatile products produced after the simulated digestion of the samples, and analyzed the lipid oxidation status of the samples. Table 1 shows the volatile unsaturated aldehydes (including the original samples) produced by the samples at various stages of digestion.

Comments 9: In line 251 of the manuscript, reference No. 30 is mentioned. Is the data here your own?

Response 9: Thank you for your kindly suggestion. In our manuscript, PCA model was used to analyze the oxidation level of the samples. This method was mentioned in reference No. 30, so we quote reference No. 30 here. At the same time, the data in the PCA model in the manuscript were provided by ourselves.

3. Additional clarifications

We have made corresponding modifications to the other questions you raised in the manuscript, please see the attachment for details. Thanks again for your comments on this article.

Reviewer 6 Report

Comments and Suggestions for Authors

The manuscript "Fatty Acid Release and Gastrointestinal Oxidation Status: 2 Different Processing Methods of Flaxseed" .

The manuscript is an original work of great practical importance for food technologists and human nutritionists. It has been prepared with great care. I consider the research methods used to be correct. In the description of the research material, I propose to complete the information on the variety of flax (botanical taxonomy) that was tested.

Author Response

1.Summary

Thank you very much for taking the time to review this manuscript. Please find the detailed responses below, along with the corresponding revisions highlighted in the resubmission.

2.Point-by-point response to Comments and Suggestions for Authors

Comments 1: In the description of the research material, I propose to complete the information on the variety of flax (botanical taxonomy) that was tested.

Response 1: Thank you for pointing this out. We have added the species of flaxseed in the text, which can be found on line 92 of the manuscript. 

Round 2

Reviewer 4 Report

Comments and Suggestions for Authors

Dear authors, thank you for taking the time to reply to my previous review. 

In Figure 3 is not clear if you compare between capital letters or capital letters to small letters for significance. Please present better.

Line 183: incorrect use of word "rules"...maybe patterns?

Lines 226-228. How can you suggest by measuring only the secondary oxidation products by GC that their oxidation level before digestion is low? How about primary oxidation before digestion?

Regarding increase or decrease in the sizes of droplets, these changes are due to eg coalesence? aggregation? Did you notice any of these phenomena?

Lines 263-265. Seconday products (aldehydes etc) are the result of hydroperoxide breakdown and b-scission reaction. When hydroperoxides break down, their concentration usually falls and secondary products increase. Therefore, these lines are not clearly written and give the wrong impression of how lipid oxidation works eg  https://doi.org/10.3390/molecules27020415  or https://core.ac.uk/download/pdf/70616398.pdf and others). The fact that you have high aldehydes does not mean necessarily that you also have high hydroperoxides. It simply means that at one point prior to the sampling you had high hydroperoxides. 

Line 335: grammar and also all the highlighted lines are from the same reference. You dont need to put it twice.

Lines 347-356. These lines are written as if they are your own results. Please rephrase

Line 367: correct spelling of whole and coupled with

In Conclusions, which process would the authors recommend after all and why?

Comments on the Quality of English Language

moderate grammar and syntax issues

Author Response

Summary

Thank you very much for taking time out of your busy schedule to review this manuscript again. Please find the detailed responses below, along with the corresponding revisions highlighted in the resubmission.

2.Point-by-point response to Comments and Suggestions for Authors

Comments 1: In Figure 3 is not clear if you compare between capital letters or capital letters to small letters for significance. Please present better.

Response 1: Thank you for pointing this out. We use upper and lower case letters to indicate data difference significance to distinguish data from different groups. We supplement this in Figure 3.

Comments 2: Line 183: incorrect use of word "rules"...maybe patterns?

Response 2: Thank you for your kindly suggestion. We have amended the word in line 183 of the manuscript.

Comments 3: Lines 226-228. How can you suggest by measuring only the secondary oxidation products by GC that their oxidation level before digestion is low? How about primary oxidation before digestion?

Response 3: Thank you for pointing this out. We tested the peroxide value of the samples before the experiment, but the obtained value did not reach the effective limit of the peroxide value. Therefore, we determined that the peroxide value and oxidation degree of the samples before digestion were low. In the manuscript, we have supplemented this.

Comments 4Regarding increase or decrease in the sizes of droplets, these changes are due to eg coalesence? aggregation? Did you notice any of these phenomena?

Response 4: Thank you for your questions. In line 248 of the manuscript, we gave a supplementary explanation of the phenomenon of particle size increase.

Comments 5: Lines 263-265. Seconday products (aldehydes etc) are the result of hydroperoxide breakdown and b-scission reaction. When hydroperoxides break down, their concentration usually falls and secondary products increase. Therefore, these lines are not clearly written and give the wrong impression of how lipid oxidation works eg  https://doi.org/10.3390/molecules27020415  or https://core.ac.uk/download/pdf/70616398.pdf and others). The fact that you have high aldehydes does not mean necessarily that you also have high hydroperoxides. It simply means that at one point prior to the sampling you had high hydroperoxides. 

Response 5: Thank you for pointing this out. We believe that in the initial stage of lipid oxidation, the concentration of hydroperoxide increases. With the increase of oxidation time, the concentration of hydroperoxide will decrease. The total concentration of volatile oxidation products will continue to rise throughout the oxidation process. Then, when the concentration of volatile products increases, the concentration of hydroperoxide may still be at a high value or has shown a downward trend. It is undeniable that the concentration of volatile products has increased because the concentration of hydroperoxides has increased. Therefore, we believe that high concentrations of volatile unsaturated aldehydes and high concentrations of hydroperoxides do not necessarily occur simultaneously. The expression of this point in our manuscript is not appropriate, and we have corrected it.

Comments 6: Line 335: grammar and also all the highlighted lines are from the same reference. You dont need to put it twice.

Lines 347-356. These lines are written as if they are your own results. Please rephrase

Line 367: correct spelling of whole and coupled with

Response 6: Thank you for your kindly suggestion. We have revised the relevant issues in the manuscript.

Comments 7: In Conclusions, which process would the authors recommend after all and why?

Response 7: Thank you for your question. We believe that all kinds of treatment methods for flaxseed have their own advantages. In terms of microwaving flaxseed and flaxseed powder, the processing methods of this kind of solid particle product is relatively simple, the cost is also lower, and it is easier to eat and store. Flaxseed plant-based milk made with a more complex treatment can help release the omega-3 PUFAs contained in flaxseed in the gastrointestinal tract. At the same time, through our study, we found that when omega-3 PUFAs were ingested through flaxseed plant-based milk, there was no significant increase in the degree of lipid oxidation of omega-3 PUFAs compared to ingesting flaxseed in solid granular form.